# Retrieval of the vertical evolution of the cloud effective radius from the Chinese FY-4 next-generation geostationary satellite

Yilun Chen[1, 2, 5], Guangcan Chen[1], Chunguang Cui[3], Aoqi Zhang[1, 2, 5], Rong Wan[3], Shengnan Zhou[1, 4], Dongyong Wang[4], Yunfei Fu[1]

[1]School of Earth and Space Sciences, University of Science and Technology of China, Hefei, 230026, China.
[2]School of Atmospheric Sciences, Sun Yat-Sen University, Zhuhai, 519082, China.
[3]Hubei Key Laboratory for Heavy Rain Monitoring and Warning Research, Institute of Heavy Rain, China Meteorological Administration, Wuhan, 430205, China.
[4]Anhui Meteorological Observatory, Hefei, 230001, China.
[5]Southern Marine Science and Engineering Guangdong Laboratory (Zhuhai), Zhuhai, 519082, China

*Correspondence to*: Yunfei Fu (fyf@ustc.edu.cn)

**Abstract.** The vertical evolution of the cloud effective radius ($R_e$) reflects the precipitation-forming process. Based on observations from the first Chinese next-generation geostationary meteorological satellites (FY-4A), we established a new method for objectively obtaining the vertical temperature vs $R_e$ profile. First of all, $R_e$ was calculated using a bi-spectrum lookup table. And then, cloud clusters were objectively identified using the maximum temperature gradient method. Finally, the $R_e$ profile in a certain cloud was then obtained by combining these two sets of data. Compared with the conventional method used to obtain the $R_e$ profile from the subjective division of a region, objective cloud cluster identification establishes a unified standard, increases the credibility of the $R_e$ profile and facilitates the comparison of different $R_e$ profiles. To investigate its performance, we selected a heavy precipitation event from the Integrative Monsoon Frontal Rainfall Experiment in summer 2018. The results showed that the method successfully identified and tracked the cloud cluster. The $R_e$ profile showed completely different morphologies in different life stages of the cloud cluster, which is important in the characterization of the formation of precipitation and the temporal evolution of microphysical processes.

## 1 Introduction

More than half of the Earth's surface is covered by clouds. As an important part of the Earth–atmosphere system, clouds affect the radiation budget through reflection, transmission, absorption and emission and therefore affect both the weather and climate (Liou, 1986; Rossow and Schiffer, 1999). Clouds also affect the water cycle through controlling precipitation, which is the main way that the water in the atmosphere returns to the surface (Oki and Kanae, 2006). Different clouds have different cloud-top heights, morphology, particle size and optical thickness (Rangno and Hobbs, 2005). Changes in the droplet size in clouds affect climate sensitivity (Wetherald and Manabe, 1988) and can also characterize the indirect effects of aerosols (Rosenfeld et al., 2007; Rosenfeld et al., 2012). An understanding of the microphysical characteristics of clouds is a prerequisite of determining their impact on the water cycle and their radiative effects on Earth's climate system.

The cloud effective radius ($R_e$) is the core parameter representing the microphysical characteristics of clouds and is closely related to the processes forming precipitation. Freud and Rosenfeld (2012) showed that the rate of droplet coalescence is proportional to the 5th power of the mean volume radius, which means that the change in the droplet coalescence rate is small when $R_e$ is small and warm rain is efficiently formed when $R_e$ is >14 μm. Similarly, for marine stratocumulus clouds, when $R_e$ is <14 μm, the column maximum rain intensity is almost <0.1 mm h$^{-1}$, but the intensity of rain increases rapidly as $R_e$ exceeds this threshold, regardless of the cloud water path (Rosenfeld et al., 2012). To date, a large number of studies have illustrated this crucial threshold using simulations, satellite remote sensing and aircraft observations (Rosenfeld and Gutman, 1994; Suzuki et al., 2010; Suzuki et al., 2011; Braga et al., 2017). The existence of this crucial threshold can also be used to explain the suppressing effect of anthropogenic aerosols on precipitation. More aerosols result in more cloud condensation nuclei (CCN) and smaller $R_e$ with coalescence occurring at a higher altitude during ascent (Rosenfeld, 1999; Rosenfeld, 2000).

The vertical evolution of $R_e$ is a fundamental property describing the development of the whole cloud cluster (Rosenfeld, 2018). There have been many studies of the vertical profiles of microphysical properties based on observations from aircraft (Andreae et al., 2004; Rosenfeld et al., 2006; Prabha et al., 2011). Pawlowska et al. (2000) showed that $R_e$ varies regularly with altitude. Painemal and Zuidema (2011) normalized the vertical profiles of microphysical properties by the cloud-top height and the in-cloud maximum value and obtained adiabatic-like profiles with the maximum value of $R_e$ near the cloud top. Wendisch et al. (2016) found that $R_e$ increases rapidly with height in clean clouds, but increases slowly in polluted regions. Although aircraft observations can intuitively obtain the vertical structure of microphysical parameters in clouds, they are limited by the platform itself and it is difficult to make continuous, wide observations. Satellite remote sensing has a global perspective that captures multiple clouds in an area at the same time.

It is difficult to directly retrieve the vertical profile of $R_e$ using satellite visible and infrared bands. By establishing the weighting functions of near-infrared atmospheric window bands, Platnick (2000) attempted to develop retrieval algorithms for $R_e$ profiles in specific clouds. Chang and Li (2002; 2003) further developed this method using multispectral near-infrared bands from Moderate-Resolution Imaging Spectroradiometer (MODIS) observations. However, their algorithm is highly sensitive to small changes in reflectance and the requirements for cloud uniformity, instrument error and model error are very high. As such, the algorithm cannot be widely applied to existing satellite observations (King and Vaughan, 2012). Recently, Ewald et al. (2019) developed an algorithm using reflected solar radiation from cloud sides, which may provide a new perspective on the vertical evolution of $R_e$.

Pioneering work by Rosenfeld and Lensky (1998) introduced a technique to correlate the change in $R_e$ with cloud-top temperature. This technique was subsequently applied to a wide range of instruments onboard polar-orbiting satellites and revealed the effects of anthropogenic aerosols on precipitation, the effects of aerosols on glaciation temperatures, the vertical profiles of microphysical properties in strongly convective clouds and the retrieval of CCN concentrations (Rosenfeld, 2000; Rosenfeld et al., 2005; Ansmann et al., 2008; Rosenfeld et al., 2008; Rosenfeld et al., 2011; Zheng and Rosenfeld, 2015; Rosenfeld, 2018). The core of this technique was to assume that the $R_e$ and temperature of the cloud top (the cloud surface observed by the satellite) were the same as the $R_e$ and temperature within the cloud at the same height and that the relationship

between $R_e$ and temperature in a given region at a given time was similar to the $R_e$–temperature time evolution of a given cloud at one location. Lensky and Rosenfeld (2006) applied this technique to observations from geostationary satellites and obtained the development and evolution of temperature and $R_e$ for several convective cells.

These studies effectively revealed the $R_e$ profiles of different clouds, but there are still some areas that require improvement, the most important of which is the selection of the study area. Previous work typically used a subjective polygon to select the study area and then calculated the $R_e$–temperature relationship in that area. For example, Rosenfeld and Lensky (1998) specified that a study should "define a window containing a convective cloud cluster with elements representing all growing stages, typically containing several thousand[s] pixels". This method is suitable for experienced scientists, but not conducive to the repeated work of other researchers. In the face of large systems (such as mesoscale convective systems), it is difficult for researchers to explain why polygons are used to frame such specific regions (the shape of the polygons and the actual clouds are clearly different). It is therefore necessary to develop an objective cloud cluster identification method and to calculate the $R_e$ vertical profile of the cloud cluster. This can solve these problems, increase the credibility of the $R_e$ profile and facilitate the comparison of $R_e$ vertical profiles in different regions. Although some active instruments (e.g. Cloud Profiling Radar) can already retrieve $R_e$ profiles effectively (Delanoe and Hogan, 2010; Deng et al., 2013), to the best of our knowledge, no passive instrument onboard geostationary satellite has yet provided an operational $R_e$ vertical profile product.

The aim of this study was to automatically identify and track the development and evolution of cloud clusters based on objective cloud cluster identification and to obtain the $R_e$ vertical profiles of these objectively identified clusters. Incorporating this technique into observations from geostationary satellites will give $R_e$ vertical profiles of a specific convective system at different life stages, helping to explain the mechanism for the formation of precipitation and changes in the upper glaciation temperature. The algorithm was applied to the first Chinese next-generation geostationary meteorological satellites (FY-4A) as a new science product.

## 2 Data and methods

### 2.1 Data

FY-4A was launched on December 11, 2016 with a longitude centered at 104.7° E (Yang et al., 2017). FY-4A data have been available since March 12, 2018 and can be downloaded from the FENGYUN Satellite Data Center (data.nsmc.org.cn). FY-4A has improved weather observations in several ways compared with the first generation of Chinese geostationary satellites (FY-2). For example, FY-4 is equipped with an Advanced Geosynchronous Radiation Imager (AGRI) with 14 spectral bands (FY-2 has five bands), with a resolution of 1 km in the visible bands (centered at 0.47, 0.65, and 0.825 μm), 2 km in the near-infrared bands (centered at 1.375, 1.61, 2.225, and 3.75 μm) and 4 km in the infrared bands (centered at 3.75, 6.25, 7.1, 8.5, 10.8, 12.0, and 13.5 μm). FY-4 AGRI provides a full-disk scan every 15 minutes (FY-2 every 30 minutes) and the scan period is shorter over China (Chinese regional scan), which helps to identify and track convective clouds. FY-4 products have been used to retrieve the cloud mask, volcanic ash height and other scientific products (Min, 2017; Zhu et al., 2017).

The introduction of the near-infrared band makes it possible to retrieve $R_e$ using FY-4 AGRI. Figure 1 shows the shortwave spectral characteristics of AGRI bands (<2 μm) in the water vapour window. We used the 0.65 and 1.61 μm channels to

establish a bi-spectrum lookup table to retrieve the cloud optical thickness (τ) and $R_e$. Both channels have a signal-to-noise ratio >200. We selected FY-4 AGRI Chinese regional scan data from June 29 to June 30, 2018. Central and eastern China experienced heavy rain during the Meiyu period at this time and the Integrative Monsoon Frontal Rainfall Experiment was underway. Figure 2 shows an example of 0.65 μm (Figure 2a), 1.61 μm (Figure 2b), and 10.8 μm (Figure 2c) channels of the AGRI and the three important parameters, including the solar zenith angle (Figure 2d), the viewing zenith angle (Figure 2e),

and the relative azimuth (Figure 2f). These six parameters were used for the retrieval of the $R_e$ profiles.

## 2.2 Methods

### 2.2.1 $R_e$ retrieval

The spectral $R_e$ retrieval algorithms, in which the bi-spectral reflectance algorithm is the most representative, are based on the optical characteristics of the cloud itself. It was first proposed by Twomey and Seton (1980) to calculate τ and $R_e$. Subsequently,

Nakajima and King (1990) extended the scope of the retrieval algorithm and constructed a lookup table, which is currently the official algorithm for MODIS cloud properties. The basic principles of the retrieval algorithm are that the absorption of the cloud droplets is negligible in the visible band and the reflectance mainly depends on the value of τ. In the near-infrared band, the reflection function mainly depends on the cloud particle radius: the smaller the radius, the greater the reflection function. This allows the simultaneous retrieval of τ and $R_e$. This method has been widely used for the retrieval of cloud properties from

multiple onboard instruments (Kawamoto et al., 2001; Fu, 2014; Letu et al., 2019).

We used libRadtran to construct a lookup table for the retrieval of cloud properties. libRadtran is a collection of C and Fortran functions and programs used to calculate solar and thermal radiation in the Earth's atmosphere (Mayer and Kylling, 2005; Emde et al., 2016). Specifically, the atmospheric molecular parameterization scheme selects the LOWTRAN scheme. For water clouds, we select Mie scheme which reads in pre-calculated Mie tables (http://www.libradtran.org). Single scattering

properties of ice clouds are obtained from Yang et al. (2013) using the severely roughened aggregated column ice crystal habit. The atmospheric temperature and humidity profiles are the preset mid-latitude summer profiles. Considering that we are mainly concerned with cloud cluster with precipitation, in order to simplify the model and speed up the calculation, we closed the aerosol module. The setting of the surface type affects the surface albedo. We currently only set the two types of underlying surfaces, mixed_forest and ocean_water. The model simulation takes full account of the spectral response functions of the FY-

4 AGRI 0.65 and 1.61 μm channels.

The lookup table is as a function of τ, $R_e$, solar zenith angle (SZA), viewing zenith angle (VZA), and the relative azimuth (AZ) between the sun and the satellite. Table 1 summarizes the range and grid points for τ, $R_e$, SZA, VZA and AZ used in constructing lookup table. Figure 3 shows an example of τ and $R_e$ over ocean_water underlying surface for water cloud and ice cloud. The dashed lines represent reflectance contours for fixed τ, and the solid lines are for fixed $R_e$. Since ice and liquid

phase clouds have different scattering properties, it is critical to classify the cloud thermodynamic phase in the retrieval process.

It is generally believed that pixels with brightness temperature lower than 233 K are covered by ice clouds, and greater than 273 K are water clouds (Menzel and Strabala, 1997), and therefore the threshold for pure ice cloud and pure water cloud is 233 K and 273 K respectively. When the brightness temperature is between 233 K and 273 K, we bring the reflectance into the water-cloud and ice-cloud lookup table simultaneously. As shown in Figure 3, some combinations of reflectance are definitely ice clouds (or water clouds), and then they are treated as pure ice clouds (or water clouds), using the corresponding retrieval lookup table. Otherwise, the differences between the brightness temperature and 233 K (and 273 K) are used as the weights, multiplied by the retrieval values from the water-cloud (and ice-cloud) lookup table, and then divide the sum of the two by 40, to obtain the cloud parameters of the mixed-cloud pixel. Considering the fact that the thermal infrared channel providing key phase information has a resolution of 4 km (much coarser than MODIS of 1 km), there may be many possibilities such as pure water cloud, pure ice cloud, mixed-phase cloud, multilayer cloud or broken cloud in the pixel. Please note that the current algorithm is difficult to handle multilayer cloud and broken cloud. The 1.61 μm channel is also affected by factors such as water vapour and $CO_2$, that is, cloud height may be sensitive to 1.61 μm reflectance. Through conducting the radiative transfer calculations under the most extreme conditions, we found that the impact of cloud height difference in reflectance would not exceed 8% (figure omitted).


2.2.2 Cloud-cluster identification

The occurrence, development and dissipation of cloud clusters results in changes in their location, area, cloud-top temperature, average temperature and precipitation. The process of these changes is relatively continuous (Zhang and Fu, 2018) and continuous pixels with a certain feature are often used to identify a "cloud cluster" or convective system (Mapes and Houze, 1993; Zuidema, 2003; Chen et al., 2017; Chen and Fu, 2017; Huang et al., 2017). For example, Williams and Houze (1987) only considered continuous areas in which the brightness temperature was <213 K when identifying and tracking cloud clusters. However, this algorithm is not suitable for the calculation of the vertical profile of $R_e$ because it only calculates the core area of the convective cloud and ignores the vast areas of low clouds. It therefore cannot obtain a complete $R_e$ profile in the vertical direction. If a higher brightness temperature threshold (e.g., 285 K) is used, it is possible to identify a cloud belt that is thousands of kilometers long (such as the Meiyu front system in China). It is not appropriate to treat such a large system as one cloud cluster.

The strong convective core of a cloud cluster appears as a low value in the brightness temperature and the surrounding brightness temperature increases as the distance from the core increases. Using this principle, we took the brightness temperature of the 10.8 μm channel and calculated the maximum gradient direction of the brightness temperature of each pixel. We then searched sequentially along this direction until the local minimum point (the cloud convection core) was reached. If this point was marked, then a number of independent cloud clusters could be identified in a large system. The specific algorithm sequence is as follows.

1) The 10.8 μm channel brightness temperature is pre-processed through a Gaussian filter with a standard deviation of 10 pixels and truncated at 4 times the standard deviation. The cloud is assumed to be inhomogeneous and the AGRI instrument

has inherent errors in the observations. This means that the final brightness temperature may change over a short horizontal distance. These changes are not physically identified as independent cloud clusters, but will affect the stability of the algorithm. Gaussian filtering can smooth out the noise of these local minima by retaining the cloud convective core.

2) The pre-processed 10.8 μm brightness temperature is used to find the local temperature minimum. The local temperature minimum represents the center of the convective core, although there may be multiple convective cores around the lowest

temperature core. These convective cores cannot be considered as independent cloud clusters in terms of short distance. Therefore, we first calculate local temperature minima for the complete scene (the brightness temperature is lower than the surrounding 8 pixels), and then set the distance threshold to 40 km (10 pixels). If the distance between two local minimums is lower than this threshold, they would be regarded as the same cloud cluster.

3) Combining the processed 10.8 μm brightness temperature and the local minimum using the maximum temperature gradient

method, a sequential search is carried out to determine the convective core to which each pixel belongs, thereby dividing the cloud clusters (Figure 4). Specifically, take the upper left cloud pixel of the pre-processed scene as the starting point, and calculate the brightness temperature gradient of it and its surrounding cloud pixels. Find the pixel that has the greatest brightness temperature gradient with the starting pixel and consider it as the next starting pixel. Repeat this calculation until the starting point is the local minimum obtained in step 2, and then the initial starting point belongs to the cloud cluster where

this local minimum is located. After traversing all the cloud pixels as starting pixel in the scene, each cloud pixel can belong to a specific local minimum, thus an objective cloud cluster identification product can be obtained.

The scatter distribution of the $R_e$ and temperature can be obtained by pairing the retrieved $R_e$ of each pixel in the cloud cluster with the 10.8 μm brightness temperature of the pixel itself. $R_e$ is sorted by the brightness temperature and the median and other percentiles of $R_e$ are calculated every 2.5 K. To eliminate the errors caused by extreme values, a sorting calculation is only

performed in temperature intervals with >30 samples. This allows us to obtain the $R_e$ profile of the cloud cluster.

## 3 Results

The Meiyu is a persistent, almost stationary precipitation process in the Yangtze River Basin in early summer and can account for almost half of the annual precipitation in this region. The cloud system along the Meiyu front usually appears as a cloud belt with a latitudinal distribution of thousands of kilometers. It is distributed in the Sichuan Basin through the middle and

lower reaches of the Yangtze River to Japan or the western Pacific Ocean. An intensive field campaign (the Integrative Monsoon Frontal Rainfall Experiment) was conducted from June to July 2018 to determine the nature of the Meiyu frontal system through satellite observations, aircraft observations and model simulations. However, the Meiyu period was short, the precipitation was weak, the rain belt was unstable, and only three Meiyu precipitation processes occurred in 2018 (atypical Meiyu year). According to the assumption of $R_e$-profile retrieval, wide temperature distribution is beneficial to gain a complete

$R_e$ profile, and therefore we selected a heavy precipitation event in the experiment to illustrate this retrieval algorithm.

Figure 5 shows that the value of τ retrieved by Terra MODIS (Cloud_Optical_Thickness_16) and the FY-4 AGRI has a good spatial consistency and there are two large centers of τ at about (113° E, 30°N) and (119° E, 29° N), where the central value of τ is >40. A cloud band with a moderate value of τ occurs in the north of the two large centers (32–33° N) where the value of τ is about 20–40. There are regions of thin cloud and clear sky between these large-τ regions. Numerically, the value of τ

retrieved by the FY-4 AGRI is close to the MODIS result when the value of τ is small and about 10% lower than the MODIS τ when the value is large.

The $R_e$ of the two instruments showed a similar spatial distribution. The value of the FY-4 AGRI $R_e$ is also close to the MODIS result (Cloud_Effective_Radius_16) when the value is small, but different when the value is large. It is about 5 μm lower than the MODIS $R_e$ when the value is about 50 μm. The MODIS shows more detail inside the cloud band than the FY-4 AGRI. For

example, near (115° E, 34° N), the MODIS $R_e$ shows multiple large-value areas, whereas the AGRI $R_e$ is less conclusive in the same area. Similarly, in the discrimination of clear sky regions, the MODIS shows a more elaborate cloud boundary and some broken cloud regions are identified in the clear sky region. This is due to the difference in resolution between the two instruments. The horizontal resolution of the MODIS products is 1 km, whereas the horizontal resolution of the AGRI is 4 km, which inevitably leads to a lack of local detail.

Because the pixel position and spatial resolution of the FY-4 AGRI are different from those of the MODIS, the pixel-by-pixel results cannot be compared directly. The probability density function of τ and $R_e$ (Figure 6) in the region shown in Figure 5 shows the similar distribution patterns of the two instruments. The values of τ both show a unimodal distribution with a peak at around 5 and then rapidly decrease. $R_e$ appears as a double peak, corresponding to water clouds and ice clouds. Some of the MODIS $R_e$ values are >40 μm (accumulated probability <10%) and the FY-4 AGRI observations do not retrieve these large

particles. The MODIS results for τ are slightly greater than the FY-4 AGRI results.

The difference shown in Figure 6 is most likely due to the partial filling effect caused by different resolutions. Chen and Fu (2017) matched the high resolution visible pixel (~2 km) to the low resolution precipitation radar pixel (~5 km) onboard Tropical Rainfall Measuring Mission satellite and found that part of the area in the precipitation pixel measured by the radar was actually clear sky. This interpretation can also be used to explain Figure 6. We suspect that isolated cirrus clouds with a

large $R_e$ value, low clouds with a small $R_e$ value and clear skies co-exist in the 4 km region (the FY-4 AGRI pixel resolution) due to the horizontal inhomogeneity of the clouds (e.g. 115° E, 34.5° N and 112° E, 34° N in Figure 5), which means that the FY-4 AGRI only retrieves cloud properties from the overall reflectance, whereas the MODIS can obtain more detailed results. Ackerman et al. (2008) reported that the resolution has a significant impact on cloud observations and care should be taken when comparing results at different resolutions.

The different sensor zenith angle of the two instruments leads to different scattering angles, which have a large effect on the retrieval of the ice cloud $R_e$. Optically thin cirrus clouds (τ < 0.3) and the transition zones between cirrus clouds and clear skies are widely distributed in the tropics and subtropics and are difficult to observe with passive optical instruments (Fu et al., 2017). A large sensor zenith angle increases the path length through the upper troposphere, which causes the signals of thin cirrus clouds that are below the limit of resolution to be aggregated (Ackerman et al., 2008). For thin cirrus clouds generated

by convective activity, the MODIS has a much better detection capability at the edge of the scan than along the center (Maddux et al., 2010). Maddux et al. (2010) used long-term composites to show that, even for the cloud product of the MODIS itself, the $\tau$ value of the nadir is greater than the $\tau$ value of the orbital boundary (~67°) by 5-10. The $R_e$ value of the ice cloud shows differences of up to 10 μm between the near-nadir and near-edge of scans over land.

The difference in resolution of the instruments leads to a difference in the retrieval results. The MODIS L3 product releases
the cloud properties on a 1° grid. To make the retrieval results comparable, we gridded the FY-4 AGRI retrievals to this resolution in the region shown in Figure 5. In the process of gridding, $R_e$ was taken as the arithmetic mean of all the cloud pixels in the grid. In view of the physical meaning of $\tau$ itself, direct arithmetic averaging without considering the pattern of distribution within the grid produced a maximum error of 20% (Chen et al., 2019). We therefore used the logarithmic mean to average $\tau$ to a 1° grid. Figure 7 shows that, regardless of the value of $\tau$ or $R_e$, the results showed a good correlation at a 1° grid
point. The correlation coefficient of $\tau$ reached 0.959 and the correlation coefficient of $R_e$ reached 0.933.

The retrieval of cloud properties based on FY-4 AGRI was carried out successfully. Figure 8 shows the clustering result from the maximum temperature gradient method. As described in Section 2.2.2, Gaussian filtering was performed on the 10.8 μm brightness temperature before clustering, which filtered out broken clouds. The area seen as clear sky in Figure 8b (white) is therefore greater than that in Figure 5. The 10.8 μm bright temperature (Figure 8a) shows that there is a convective center
consisting of three relatively close convective cells near (113° E, 30° N) and the minimum brightness temperature is <200 K. The convective center extends to the southwest as a slender cloud band, which is consistent with the conveyor belt of water vapor during the Meiyu season. The eastern side of the convective center shows another distinct mesoscale convective system with a center at (119.5° E, 29° N) and a minimum brightness temperature of about 210 K. There is a cloud band with a brightness temperature ranging from 220 to 260 K in the north of the two main convective clouds. There are many small-scale
clouds in the north of this cloud belt.

The results of automatic clustering are consistent with subjective cognition, showing two main convective cloud clusters and several small cloud clusters on the north side. Our focus is on the convective cloud cluster on the southwest of the figure (the purple cloud cluster in Figure 8b), which produced the heaviest precipitation in the Integrative Monsoon Frontal Rainfall Experiment. The lightning generated by this cloud even destroyed some ground-based instruments. The $R_e$ profile is shown in
Figure 8d, where each red dot corresponds to the pixel-by-pixel retrieval of $R_e$ in the cloud cluster and the black line is the median value of $R_e$.

Geostationary satellites enable the continuous observation of the same area, which helps to identify and track the occurrence, development and dissipation of cloud clusters. Zhang and Fu (2018) proposed that life stage of clouds affect the convection ratio, the precipitation area, the vertical structure and characteristics of precipitation droplets. Using the FY-4 AGRI
observations, we achieved the objective segmentation of the cloud and brought the segmentation result (cloud cluster) into the continuous observations to automatically track the cloud clusters. Figure 9 tracks the purple cloud cluster shown in Figure 7. From the perspective of the brightness temperature, there were three adjacent cells with a low temperature on the west side at 00:30 UTC and a large low-temperature zone on the east side. The pattern of cells was irregularly and they were randomly

embedded in cloud bands (initiation). By 03:30 UTC, the three cells on the west side had merged to form one cloud cluster (black line), whereas the convective clouds on the east side had gradually dissipated. The convective core appeared to be a linear shape, accompanied with a large area of southwestern trailing cloud (mature). At 05:30 UTC, the cloud cluster on the west side had started to dissipate and a slender arcus cloud developed on the eastern boundary with a minimum brightness temperature <200 K. The heavy precipitation of the cloud cluster on the west side (black line) may have caused a local downburst. These cold airflows sink to the ground and flow out to the boundary of the cloud, forming a localized area of ascent with the strong solar heating in the afternoon (~13:30 local time). This closed circulation created a new, strongly convective cloud at the original cloud boundary. The original convective cloud cluster dissipated and the newly formed convective cloud cluster on the east side gradually developed and matured. From the perspective of the tracked cloud (black line), our objective tracking results successfully described the development and dissipation of this cloud cluster without confusing it with the newly generated convective cloud cluster in the east.

Figure 10 shows the evolution of the $R_e$ vertical profile for the automatically tracked cloud cluster. In terms of the area of the cluster, the rapid growth and development period of the cloud cluster was from 00:30 to 02:30 UTC. The cloud cluster area was relatively stable from 02:30 to 06:30 UTC, after which time the area was slightly decreased. In agreement with the theory of Rosenfeld and Lensky (1998), the change in $R_e$ with temperature can be divided into five distinct zones: the diffusional droplet growth zone; the droplet coalescence growth zone; the rainout zone; the mixed phase zone; and the glaciated zone. These five areas do not all necessarily appear in a given cloud cluster.

Only the small convective cell was identified at 00:30 UTC and it did not contain enough pixels with high temperature (<30 samples in 2.5-K temperature interval). Although there must be a region for particles to condense and coalesce within this cloud cluster, due to technical limitations, only glaciated zone was shown in the $R_e$ profile (Figure 10a). At 01:30 UTC, convective cells merged and the convection activity was strong. The $R_e$ profile showed that, in areas where the temperature was <230 K, $R_e$ was stable in the glaciated zone from 28 to 32 μm. $R_e$ changed almost linearly with temperature between 285 and 230 K and did not exhibit a clear boundary between the diffusional growth zone, the coalescence growth zone, the rainout zone and the mixed phase zone. This is because strong convective core usually has a strong ascending motion. Under the influence of such strong ascending motion, the boundary between the zones is broken and there is not enough time for the growth of precipitation (Figure 10b). Rosenfeld et al. (2008) explained that this situation may delay the development of both the mixed and ice phases at higher altitudes and that the resulting linear $R_e$ profile is a warning of severe weather.

The $R_e$ profile gradually showed multiple zones from 02:30 to 03:30 UTC, and the multiple zones are most distinct from 04:30 to 05:30. The median $R_e$ slowly increased with temperature from 10 to 16 μm (~285 K to 270 K), which is the growth zone in which cloud droplets are mainly condensed and is affected by the number of CCN. The growth rate of $R_e$ accelerated significantly from 16 to 22 μm (~270 K to 265 K). At this time, raindrops were formed and the growth of cloud droplets mainly depended on coalescence. The distinct zones can also be seen in the 25 and 75 percentiles, while the turning points of $R_e$ and the growth rates differ from the median. A possible explanation is that turning points and growth rates are affected not only by temperature, but also by the size of $R_e$. The rainout zone usually appears in marine cloud systems with fewer CCN (Martins et

al., 2011), whereas this precipitation process was located in inland China. A large number of artificial aerosols act as CCN to suppress warm rain while delaying freezing, which requires lower temperatures. From ~265 to ~230 K, the rate of increase in $R_e$ slightly slows down. The cloud particles gradually change from the liquid phase to the ice phase and their radius increases and absorbs more near-infrared radiation (mixed phase zone). $R_e$ remains stable below 230 K and the profile is completely in the glaciated zone. After 06:30 UTC, the intensity of the original cloud cluster was significantly weakened and gradually dissipated. The $R_e$ profile gradually became difficult to describe. This is due to the weakening of convective activity, the cloud cluster is mainly governed by a thinning anvil. Affected by the low-level clouds and high-level anvils, the uncertainty of $R_e$ retrieval increases, and the bright temperature cannot represent the cloud-top temperature at this time, which reduces the credibility of the $R_e$ profile.

## 4 Conclusions

FY-4A is the first Chinese next-generation geostationary meteorological satellites. It was launched in 2016 and began operation in 2018. Here, bi-spectral reflectance algorithm was used to retrieve $R_e$ and τ. We used the maximum temperature gradient method to automatically segment, identify and track cloud clusters. We obtained the objective cloud cluster $R_e$ profile retrieval method based on FY-4 AGRI observations by combining these two methods. Taking a severe weather event during the Integrative Monsoon Frontal Rainfall Experiment campaign as an example, we calculated the $R_e$ profiles of an objective cloud cluster at different life stages.

The cloud properties of $R_e$ and τ retrieved from the FY-4 AGRI were compared with the Terra MODIS cloud products. The results showed that they were in good agreement with the spatial distribution, although there were some differences when the value was large, which may be due to the difference in resolution and the viewing zenith angles. The results showed a strong correlation when the FY-4 AGRI and MODIS retrievals were both averaged to a 1° grid. This indicates that the cloud properties retrieved by the FY-4 AGRI were credible.

The maximum temperature gradient method effectively divides thousands of kilometers of cloud bands into multiple cloud clusters and the objective results are consistent with subjective cognition. For this specific severe weather event, the method tracked the complete process of development, maturation and dissipation of a convective cloud cluster. The $R_e$ profiles of the cloud cluster showed completely different characteristics at different life stages. During the development stage, $R_e$ changed almost linearly with temperature, whereas during the mature stage the $R_e$ profile showed multiple zones of changes with temperature. Different $R_e$ profiles reflected the different physical processes of cloud particle growth and corresponded to completely different processes of formation of precipitation.

The use of geostationary satellites to obtain continuous cloud cluster $R_e$ vertical profiles has led to many different applications. For example, the $R_e$ profile of the development stage is linear, which may help to improve the predictive skill for the nowcasting of storms. Real-time changes in the shape of the $R_e$ profile may also be used to characterize the life stages of clouds. The position of the glaciation temperature and the mixed phase zone in the $R_e$ profile indicates the formation of mixed-layer

precipitation. The continuous change in the glaciation temperature helps our understanding of mixed-layer precipitation. We are confident that the introduction of the cloud cluster $R_e$ profile will help to improve the future application of FY-4 data in meteorology.

The authors declare that they have no conflict of interest.


*Data availability.* Data supporting this paper can be found at http://www.nsmc.org.cn.

*Author contributions.* Conceptualization: YC; Investigation: YC, GC and AZ; Methodology: YC, GC and AZ; Writing: YC; Validation & Discussion & Editing: CC, RW, SZ, DW and YF; Supervision: YF.


**Acknowledgment**

This work is supported by the National Natural Science Foundation of China (Grant 91837310, 41675041, 41620104009), National Key Research and Development Program of China (Grant No. 2017YFC1501402 and 2018YFC1507200) and Key research and development projects in Anhui province (201904a07020099).

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

**Table 1. Grid point values of the lookup table parameters.**

| Quantity | # of points | Grid point values |
|---|---|---|
| $\tau$ | 34 | 0.05, 0.10, 0.25, 0.5, 0.75, 1.0, 1.25, 1.5, 1.75, 2.0, 2.39, 2.87, 3.45, 4.14, 4.97, 6.0, 7.15, 8.58, 10.30, 12.36, 14.83, 17.80, 21.36, 25.63, 30.76, 36.91, 44.30, 53.16, 63.80, 76.56, 91.88,110.26,132.31,158.78 |
| $R_e$ (µm) | 15 | 4,5,6,7,8,9,10,12,14,16,18,20,22,24,25 (liquid water cloud) |
| | 18 | 5,8,11,14,17,21,24,27,30,33,36,39,42,45,48,53,57,60 (ice cloud) |
| SZA(°) | 17 | [0, 80] equally spaced with increments of 5° |
| VZA (°) | 17 | [0,80] equally spaced with increments of 5° |
| ZA(°) | 19 | [0, 180] equally spaced with increments of 10° |


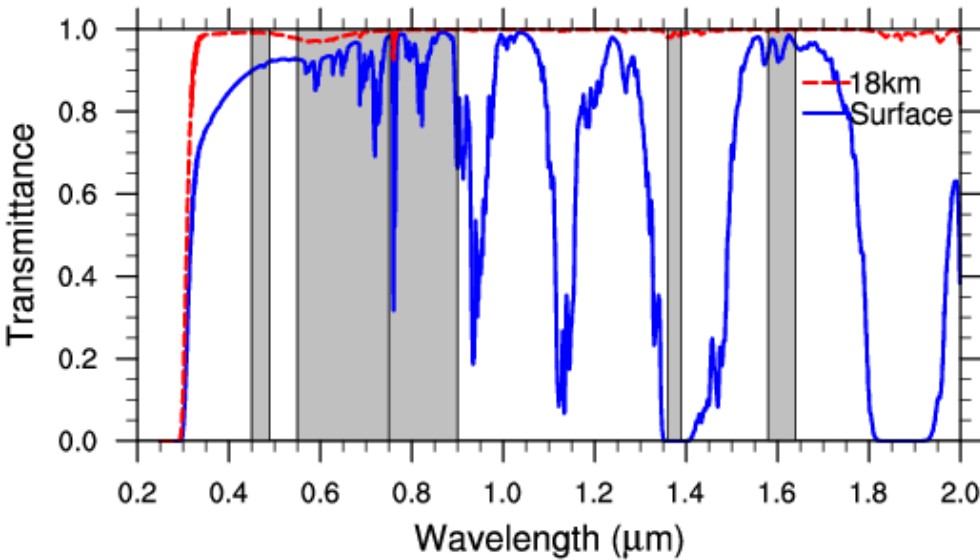

**Figure 1: Spectral characteristics of FY-4 AGRI bands centered at 0.47, 0.65, 0.825, 1.375 and 1.61 μm. The atmospheric transmittance is calculated for the mid-latitude summer temperature and humidity profiles at a solar zenith angle of 10°.**


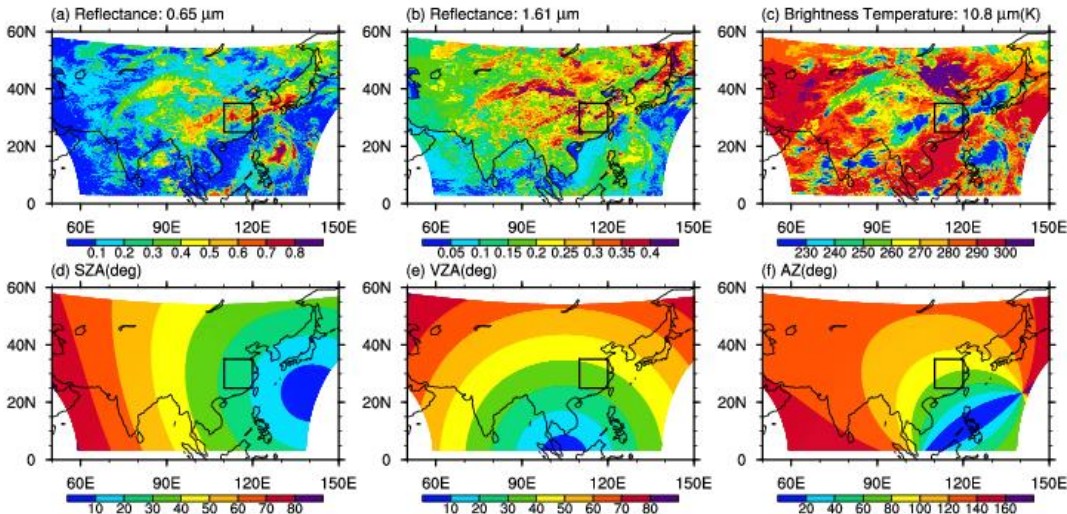

**Figure 2: Chinese regional scans and geolocation results of FY-4 AGRI at 02:38 (UTC) on June 30, 2018. (a) Reflectance at 0.65 μm; (b) reflectance at 1.61 μm; (c) brightness temperature at 10.8 μm; (d) solar zenith angle (SZA); (e) viewing zenith angle (VZA); and (f) relative azimuth (AZ). The domain shown in Figure 4 is indicated.**


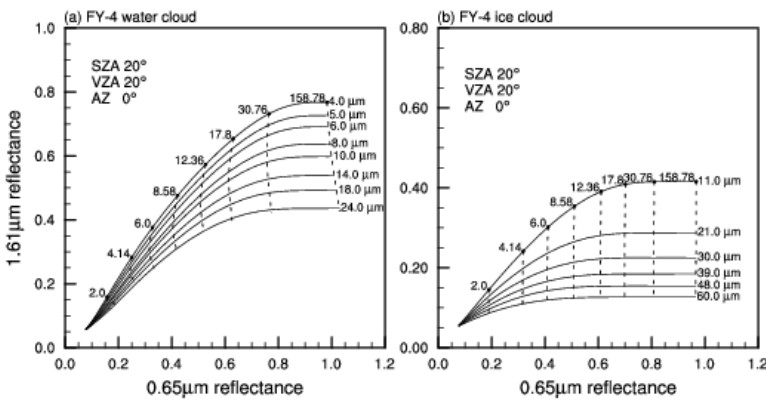

**Figure 3: Bi-spectral reflectance lookup table for FY-4 AGRI. Here, solar zenith angle=20°, viewing zenith angle=20°, relative azimuth=0°, and underlying surface=ocean_water.**

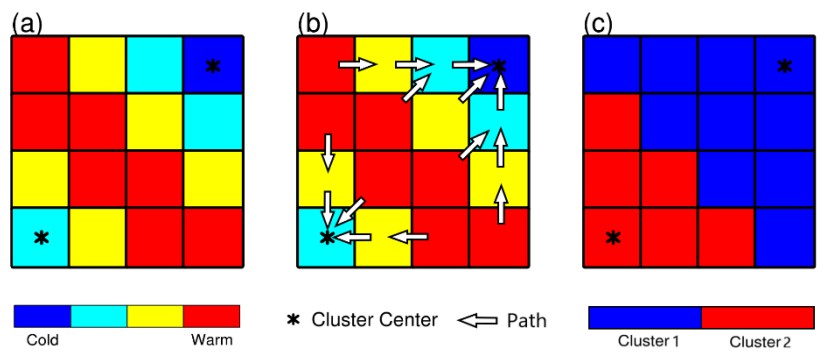


**Figure 4: schematic diagram for the maximum temperature gradient method. (a) distribution of the brightness temperature; (b) maximum brightness temperature gradient path of each cloud pixel; (c) the objective cloud cluster identification product. Please note that the local temperature minimums (asterisks) in the figure is only used to illustrate the maximum temperature gradient method. In actual calculation, the distance between the two local temperature minimums is greater than 40 km (~10 pixels).**


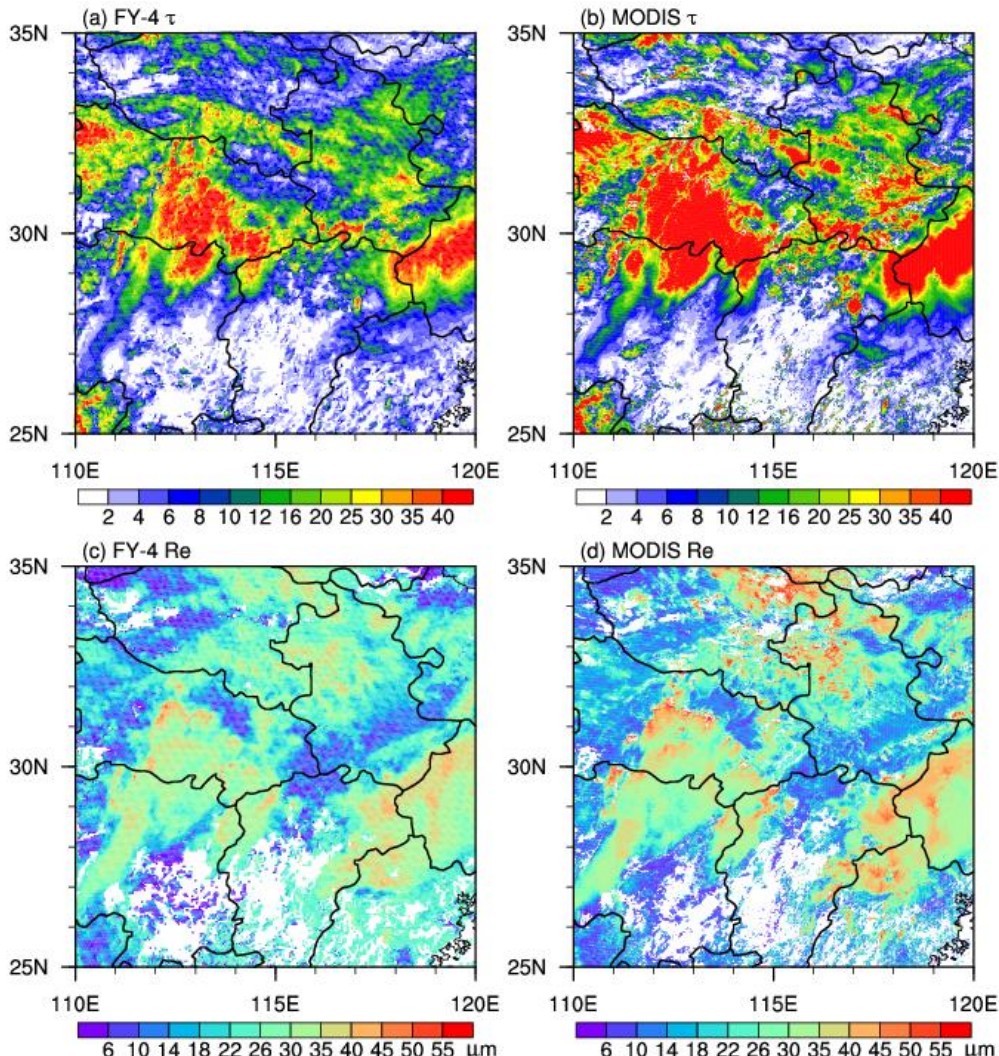

**Figure 5: Comparison of the pixel-level retrieval of cloud properties using the FY-4 AGRI with Terra MODIS cloud products (Cloud_Optical_Thickness_16 and Cloud_Effective_Radius_16). The observation time of the FY-4 AGRI is 02:38 (UTC) on June 30, 2018 and the MODIS observation time is 02:55 (UTC). The solid lines are provincial boundaries.**


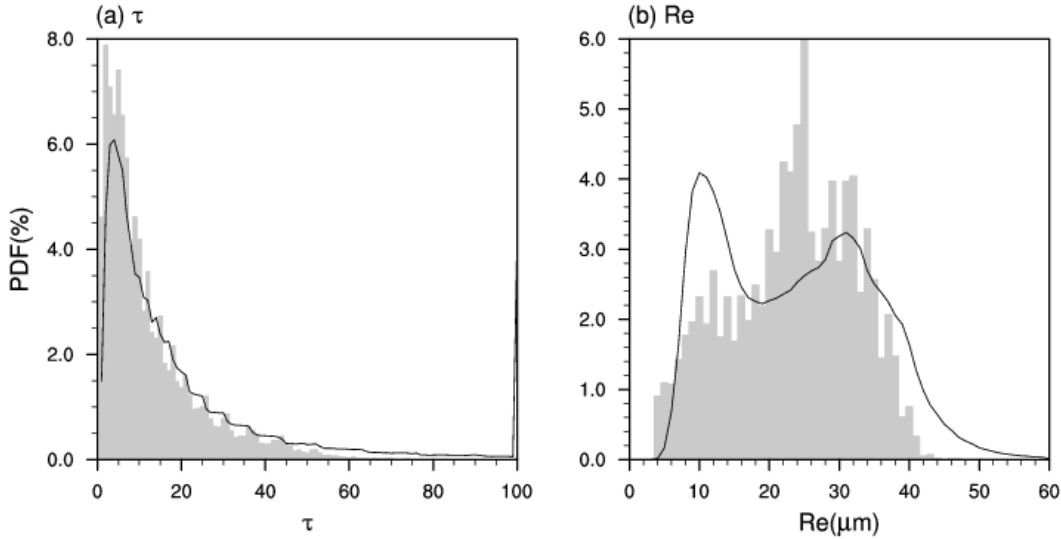

**Figure 6: Probability density function of the FY-4 AGRI retrieval results and the MODIS cloud products in the region shown in Figure 5. The shaded area shows the FY-4 AGRI results and the solid line is the MODIS results.**


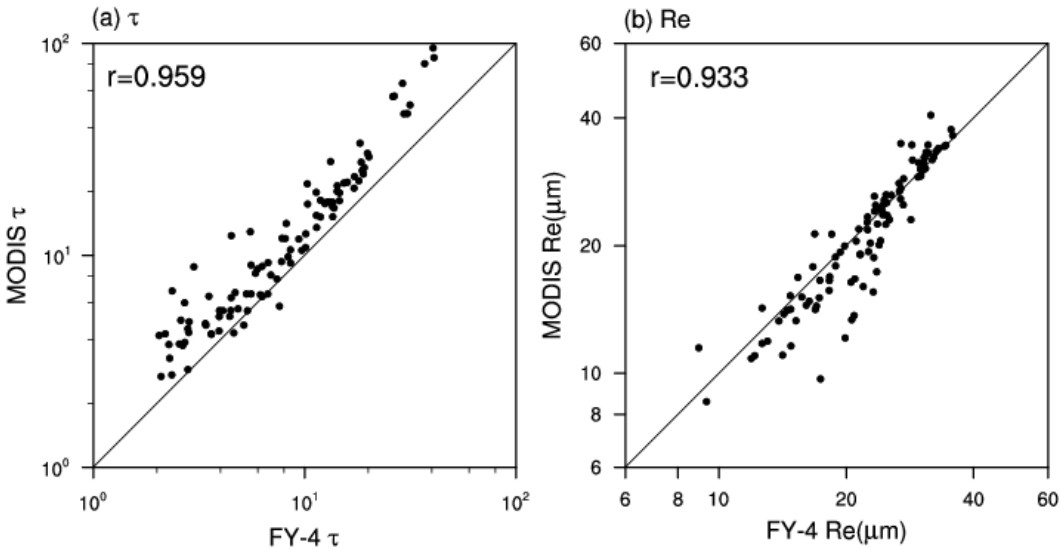

**Figure 7: Scatter plots of FY-4 AGRI and MODIS retrievals after averaging to a 1° grid in the region shown in Figure 5.**

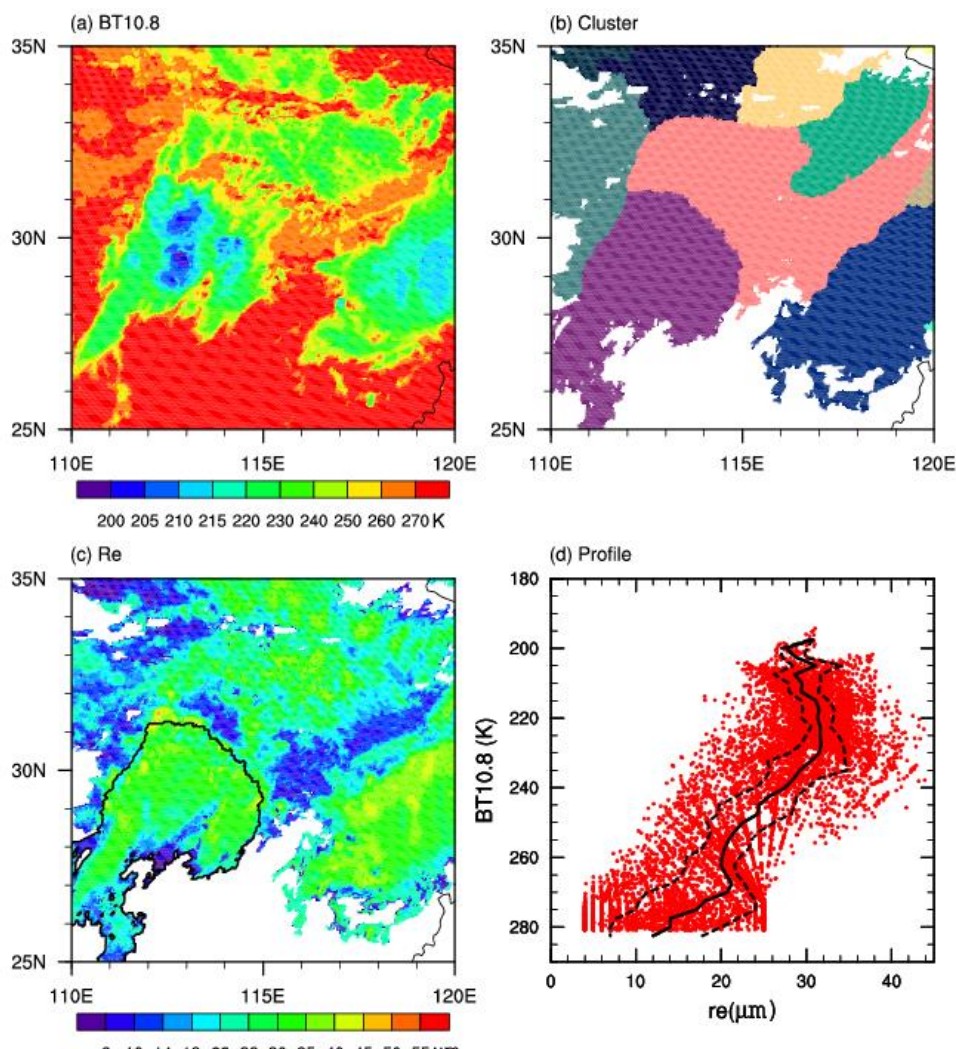

**Figure 8: Cloud cluster identification and the corresponding *Re* profile for the FY-4 AGRI observations in Figure 5 at 02:38 UTC on June 30, 2018. (a) 10.8 μm brightness temperature; (b) cloud cluster identification; (c) a specific cloud cluster identified by our algorithm with a base map of *Re*; and (d) *Re* profile of the specific cloud cluster.**

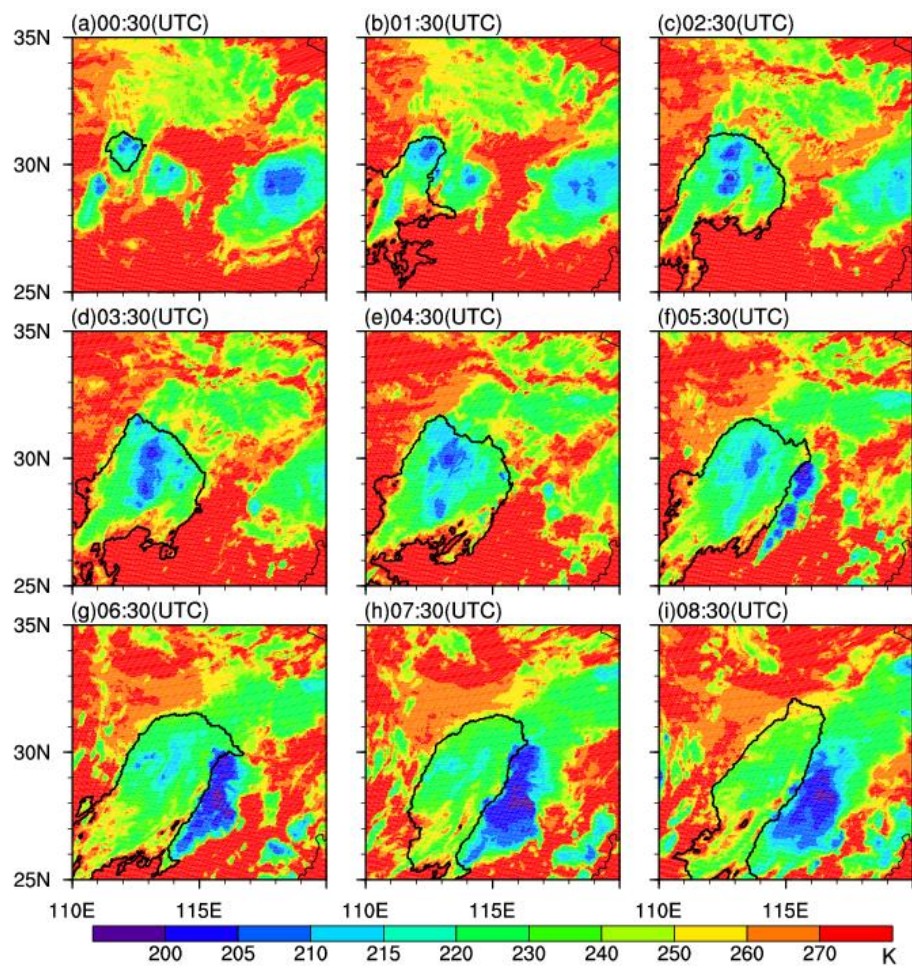

**Figure 9: Cloud cluster tracking on June 30, 2018 (one hour intervals). The black line is the continuous cloud cluster identified by the maximum temperature gradient method.**

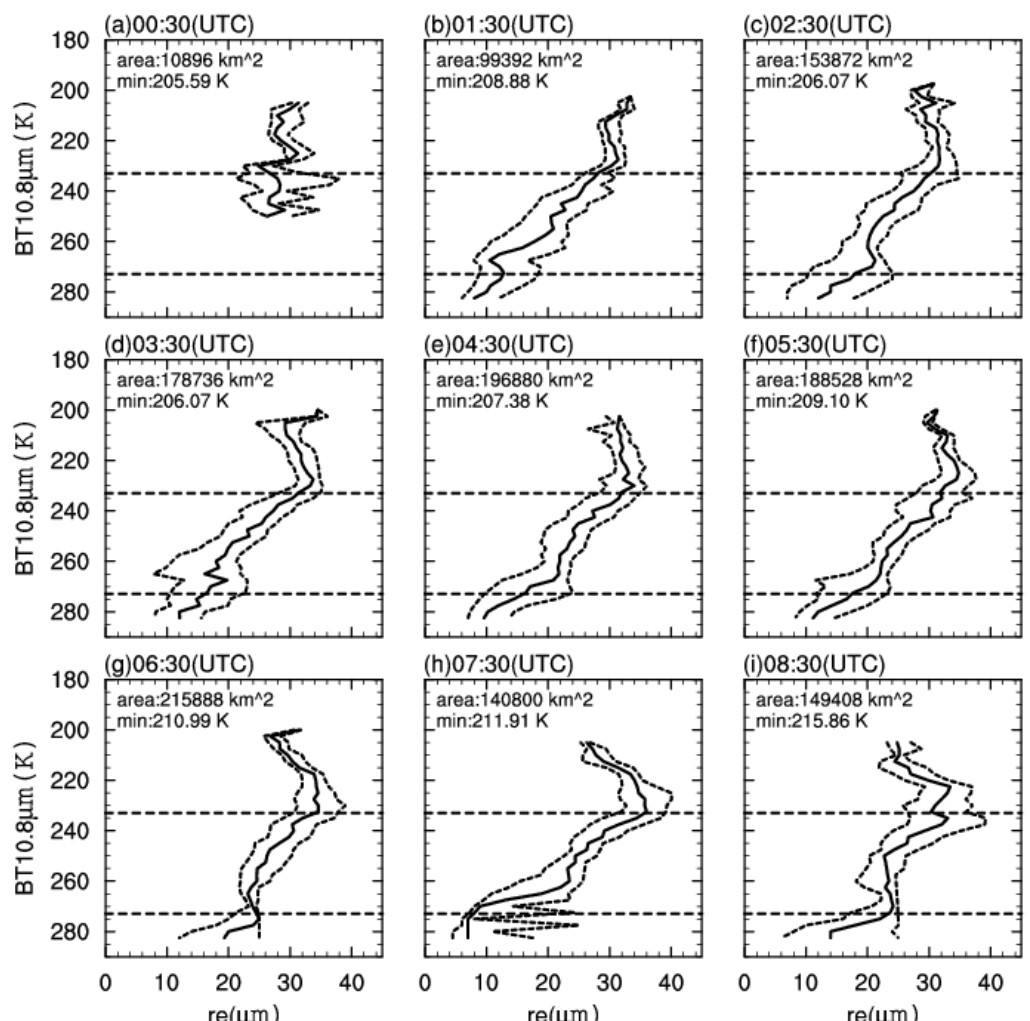

**Figure 10: Changes in the $R_e$ profile (25, 50, and 75 percentiles) in the tracked continuous cloud cluster at one hour intervals. Horizontal dashed lines represent temperatures of 273 K and 233 K, respectively. The text in the figure gives the area of the cloud cluster and the coldest 10% brightness temperature of the cluster.**
