# Peer review of "Retrieval of the vertical evolution of the cloud effective radius from the Chinese FY-4 next-generation geostationary satellite"

_Atmospheric Chemistry and Physics, 2019_

## Referee Comment (RC1) · Anonymous Referee #1 · 10 Oct 2019

Review of "retrieval of the vertical profile of the cloud effective radius from the Chinese FY-4 next-generation geostationary satellite" by Chen et al.

This manuscript studies the vertical profiles of Re in a convective cluster using the Chinese FY-4 satellite data. An objective method is used to identify the cloud cluster. Then the temperature-Re relationship is obtained for the identified cloud cluster. The temperature-Re relationship is very useful for understanding the cloud microphysical processes that produce precipitation. The method and the results are well presented in this manuscript. But it needs some revision in order to be accepted in ACP.

Major points:

[Figure]

(1) I strongly suggest that the manuscript present a figure to show how the bi-spectral reflectance vary with cloud optical depth and effective radius. The bi-spectral reflectance algorithm has been widely used (for example, in the cloud retrieval of MODIS). It would be very helpful for the readers to see how sensitive the bi-spectral reflectance is to the cloud optical depth and effective radius for the FY-4 bands. Especially this study shows that the effective radius retrieved from FY-4 is generally smaller than that retrieved from MODIS. I'm curious if this could be caused by some errors in the measurement of the near-infrared reflectance. Of course, the discrepancy between the FY-4 and MODIS retrievals may be caused by other reasons, such as different resolution, different view angles, etc.. I can understand there is discrepancy between instruments. But a figure showing the bi-spectral reflectance as a function of optical depth and effective radius for the FY-4 would be very helpful. The readers could even compare this figure with MODIS.

(2) In Figure 8, I guess it's the median radius that is shown in the figure? I suggest other percentiles, such as the 25 and 75 percentiles, should also be shown in the figure. It seems that the manuscript focuses on some subtle details in Figure 8. For example, it is said in the manuscript that 15-22 microns (03:30-05:30 UTC), correspond to a significantly accelerated growth. But this profile is only the median effective radius. I would like to see if this accelerated growth is still seen in the 25 and 75 percentiles. Similarly, 25 and 75 percentiles should also be plotted in Figure 3.

(3) The discussions related to the microphysical processes on page 8 are not very clear. I don't understand why the cloud in an earlier stage (00:30 UTC) is totally glaciated. The manuscript should at least provide some description of the convection at this stage. For temperature lower than 273 K, the cloud starts to become mixed-phase, so ice-related processes could be very important. But the manuscript seems to emphasize on the collision-coalescence process to explain the accelerated growth for the 15-22 microns (03:30-05:30 UTC). In addition, in lines 250-251, "the rate of increase in Re slows down". Why? I would expect that, in the mixed-phase, the cloud

particles are easier to get larger. Line 239-240, "did not exhibit the characteristics of the earlier zones", what does this mean?

Minor points:

(1) In the abstract, lines 13-15: I think these two sentences should be reorganized. Identifying cloud cluster is one task of this study. Obtaining the Re profiles is the other task of this study. The two sentences in the manuscript seem to mix the two tasks together.

(2) In the introduction: because the Re profile is very useful for studying precipitation formation, the first paragraph of the introduction should have some writings on precipitation. In the first sentence, only the radiation budget is mentioned. I think it should also be pointed out that clouds control precipitation, and therefore the water cycle.

(3) Line 31: the coalescence of cloud droplets is proportional to Reˆ5. What does this mean? What property of coalescence is proportional to Reˆ5?

(4) Line 76: the aim of this study was to automatically identify and . . .Therefore I think section 2.2 should be reorganized. The method of identifying the cloud cluster should first be presented, and then the method for obtaining profiles of effective radius is presented. But I'll leave it to the authors to decide on this.

(5) In section 2.1: it's better to show the total 14 band wavelength. How many bands are in the visible? How many in the near-infrared? And how many in the infraredïij§

(6) Figure 2: Figure 2 should only plots the north hemisphere. There's no need to plot the southern hemisphere in the figure. The domain shown in Figure 3 should also be indicated in Figure 2. The angles are shown in Figure 2. But I don't see any discussion of the angles in the text.

(7) Line 150-152: why is the most extreme precipitation event selected for this study? Please provide some motivation. Intuitively I would expect that this study could start with a normal precipitation case.

---

## Referee Comment (RC2) · Anonymous Referee #2 · 1 Nov 2019

**General remarks**

This is a review of the manuscript "Retrieval of the vertical profile of the cloud effective radius from the Chinese FY-4 next-generation geostationary satellite" submitted by Yilun Chen et al. to Atmospheric Chemistry and Physics Discussions.

In this manuscript, the authors present methods to study the vertical evolution of the cloud effective radius of convective cloud ensembles using data from the Chinese FY-4 next-generation geostationary satellite. The authors present a method to identify contiguous clusters of convective clouds. In a second step, a bi-spectral algorithm is

presented and used to retrieve cloud effective radius $r_{\text{eff}}$ and optical thickness $\tau$. The application of both methods is demonstrated for a heavy convective rain event where the temperature - $r_{\text{eff}}$ relationship is discussed.

I enjoyed reading this manuscript since it is mostly well written and clearly structured. The presented appraoch is the consistent further development of the ideas of Rosenfeld et al. (2008) and as such an important contribution to advance the scientific understanding of precipitation forming processes. While this work definitely deserves to be published, the description of the used methods is, however, insufficient to reproduce their results. Moreover, the presented manuscript would better fit into the scope of "Atmospheric Measurement Methods", since this manuscript is mainly about a "Retrieval of the vertical profile of the cloud effective radius". However, this decision should be made by the editor as retrieval papers can also be found in ACP. Below, I compiled a list of comments which should be considered in a revised version of this manuscript.

**Major comments**

1. **Description of methods**

   A paper in AMT/ACP should enable the reader to understand and to replicate the presented results and should not limit itself to report on its scientific advancements. The two methods central to this manuscript, however, are not adequately described to guarantee the reproducibility of the presented results. While the description of a bi-spectral retrieval algorithm should be well established and straightforward, the method to find independent cloud clusters seems new and worth to be described in more detail. In the following, I will try to give more specific advices what is still missing in section "2.2. Methods" and how to organize it in subsections:

   - First, I strongly suggest to explain the forward simulations with libRadtran in

more detail. The authors should clearly state all cloud parameters and theirs boundaries which have been varied during the forward simulations. To understand the discrepancies between retrieved $r_{\text{eff}}$ from FY-4 and MODIS, the reader needs to know the range and steps of optical thickness, effective radius, illumination and viewing angles. The current manuscript does not explain how the model clouds were set up, if an standard aerosol environment was considered and if a variable ground albedo was taken into account? The given citation for the used optical properties parameterization for ice clouds (Baum et al., 2014) also does not explain if the *baum v36* (Heymsfield et al., 2013; Yang et al., 1993; Baum et al., 2014) parameterization with the *general habit mixture*, the *rough-aggregates* or the *solid-columns* option has been used? Moreover, the authors state that they used the optical properties parameterization for liquid clouds of Hu et al. (1993), but should be aware that the developers of libRadtran state that:

*Note that this parameterization has been developed to calculate irradiances, hence it is less suitable for radiances. This is due to the use of the Henyey-Greenstein phase function as an approximation of the real Mie phase function.*

- Second, the authors do not describe how they handle the phase discrimination between water and ice clouds at all. The example discussed in section 4 clearly contains water as well as ice clouds. Inferring from the retrieved $r_{\text{eff}}$, the retrieval seems to handle water and ice clouds quite well. There is, however, no explanation if a threshold technique is used to separate ice from water clouds and how the mixed-phase region is handled. This is especially important since the discussion in section 4 is focused on the region of cloud glaciation.

- At last, section 2.2 "Methods" also introduces the technique to identify cloud clusters which is a central aspect of this study. While this technique should get its own subsection, it also deserves a more visual and complete description: The authors miss to provide important details, like the size of the Gaussian filter and the used value of the distance threshold. Furthermore, the authors write that "**the** local temperature minimum" (P5, L135) is determined, but to the reader it is not clear if this is done pixel-wise or for the complete scene. Do you identify all local temperature minima in the scene or only for a search radius around each pixel?

Moreover, the authors write on P5, L139ff:

*"3) Combining the processed 10.8 micron brightness temperature and the local minimum using **the maximum temperature gradient method**, a sequential search is carried out to determine the convective core to which each pixel belongs, thereby dividing the cloud clusters."*

Here, it is not clear how *the maximum temperature gradient method* (which is never explained!) can be combined with a brightness temperature to determine the convective core for each pixel in a *sequential search* (which is also not explained). Here, a descriptive figure could significantly improve the comprehensibility of this paragraph. In my opinion, the revision of this section should be of major concern since it seems to be the main novelty of this work.

**2. Discussion of results**

As also pointed out by RC1, the discussion about the microphysical evolution of the cloud cluster on page 8 is not very convincing. In my opinion, the authors focus on details in Figure 8 and on processes (collision-coalescence, precipitation formation), which their spaceborne technique probably never can resolve in detail. As long as the handling or the influence of mixed-phase cloud regions is not explained, their discussion oversells their approach while it misses to highlight its strength: to observe the timescales between initiation, invigoration and the mature phase of a convective cluster. Examples of unclear or unproven sentences are:

Page 8, L239f: *"Re changed almost linearly with temperature between 285 and 230 K and did not exhibit the characteristics of the earlier zones"*

What do you mean here by "earlier zones"?

Page 8, L241f: *"Under the influence of such strong ascending motion, the boundary between the zones is broken and there is not enough time for the growth of precipitation."*

This statement is incomprehensible to me since I can not observe clear boundaries in Figure 8.

Page 8, L255f: *"In addition, because of the deposition of aerosols after precipitation, sufficient water vapor allowed Re to exceed $20\,\mu m$ at higher temperatures."*

Can you deduce this observation from your retrieval results alone? I doubt that you can observe the deposition of aerosols from a geostationary satellite. As the cluster during this phase is mainly governed by a thinning anvil, multilayer cloud effects have to be taken into account for the discussion of the observed $r_{\text{eff}}$ profile.

**Minor comments**

- **Title:** I suggest a slight change to the title of the manuscript, since the original title "Retrieval of the vertical profile ... " gives the impression of a retrieval which can be applied to a single cloud like multi-wavelength retrievals (e.g. Chang et al. (2003)). In my opinion, the title "Retrieval of the vertical evolution of the cloud effective radius from the Chinese FY-4 next-generation geostationary satellite"

better captures the approach to retrieve the vertical profile of $r_{\mathrm{eff}}$ by observing the evolution of $r_{\mathrm{eff}}$ using a cloud ensemble approach.

- **References:** Please check all references for chronological order

- **P1, L31:** Freud and Rosenfeld (2003) showed that the rate of droplet coalescence is proportional to the mean volume radius $r_v^5$ and not the mean effective radius $r_{\mathrm{eff}}^5$.

- **P2, L38f:** Reword you sentence "More aerosols result in more cloud condensation nuclei (CCN), leading to a higher height of the $14\,\mu m$ threshold for Re and a smaller coalescence efficiency" into "More aerosols result in more cloud condensation nuclei (CCN) and smaller $r_{\mathrm{eff}}$ with coalescence occurring at an higher altitude during ascent"

- **P2, L49f:** Besides the multi-wavelength approach you should also mention the cloud side perspective approach to directly retrieve the vertical profile of $r_{\mathrm{eff}}$ (e.g. Ewald et al. (2019)).

- **P3, L74f:** *"To the best of our knowledge, no instrument has yet provided an official Re vertical profile product."*

This statement is not true. You should at least mention multi-instrument products like DARDAR, 2C-ICE or Cloudnet which provide effective radius profiles on an operational basis:

*Delanoe, J., and R. J. Hogan, 2010: Combined CloudSat-CALIPSO-MODIS retrievals of the properties of ice clouds. J. Geophys. Res., 115, D00H29.*

*Deng M, Gerald G. Mace, Zhien Wang, and R. Paul Lawson, 2013: Evaluation of Several A-Train Ice Cloud Retrieval Products with In Situ Measurements Collected during the SPARTICUS Campaign. J. Appl. Meteor. Climatol., 52, 1014–1030.*

Moreover, it is not clear what you mean with "official". Maybe you mean "operational" in this context?

- **P4, L95** "We selected Chinese regional data". Please be more precise what data and from which source (model, measurements?).

- **P4, L98** Please refer to different sub-panels (a, b, c) in Figure 2. Moreover I do not understand what you mean with "closely related to the retrieval".

- **P4, L100** *"The spectral retrieval algorithm of cloud properties is based on the characteristics of the cloud itself and the bi-spectral reflectance algorithm is the most representative."*

  This sentence is incomprehensible. What do you mean by that?

- **P5, L130** Please elaborate what you mean by "The original data are pre-processed".

- **P5, L154ff** You only mention MODIS in your manuscript and never mention the satellite and the actual retrieval product you used in your comparison. You obviously used data from MODIS on Terra. Moreover, there are multiple scientific datasets (SDS) for cloud effective radius retrieved with different techniques and filters. Did you use the SDS *Cloud_Effective_Radius_16* with the same channel combination?

- **P6, L174** Weather radars (here with a coarser resolution than the satellite!) should not be used to explain resolution effects between different satellites working in the visible wavelength region. Drizzle or a few rain drops in a pixel can give you a radar signal which seems to be clear in the visible wavelength region.

- **P9, L274** "The glaciation temperature increased significantly during the period of dissipation" Have you shown this observation in the results? And with which method?

**Wording**

- **P1, L2** "from the first of the Chinese" ... "from the first Chinese"

- **P1, L28** "determining the effects of radiation and the water cycle on the Earth's climate system" ... "determining their impact on the water cycle and their radiative effects on Earth's climate system"

- **P2, L55** "obtain" ... "correlate"

- **P3, L93** "the shortwave distribution of AGRI" ... "the shortwave spectral characteristics of AGRI bands"

- **P4, L97** Please rephrase "... and the three angles important for retrieval"

- **P9, L259** "We used bi-spectral reflectance observations from the FY-4 AGRI to calculate a lookup table to retrieve Re and $\tau$." This sentence does not make sense.

**References**

Baum, B. A., Yang, P., Heymsfield, A. J., Bansemer, A., Merrelli, A., Schmitt, C., and Wang, C.: Ice cloud bulk single-scattering property models with the full phase matrix at wavelengths from 0.2 to 100 mu, J. Quant. Spectrosc. Radiat. Transfer, special Issue ELS-XIV, 2014.

Chang, F.-L., and Li, Z. (2003), Retrieving vertical profiles of water-cloud droplet effective radius: Algorithm modification and preliminary application, J. Geophys. Res., 108, 4763, doi:10.1029/2003JD003906, D24.

Delanoe, J., Hogan, R., et al.: Combined CloudSat-CALIPSO-MODIS retrievals of the properties of ice clouds, Journal of Geophysical Research, 115, 2010.

Ewald, F., Zinner, T., Kölling, T., and Mayer, B.: Remote sensing of cloud droplet radius profiles using solar reflectance from cloud sides – Part 1: Retrieval development and characterization, Atmos. Meas. Tech., 12, 1183–1206, https://doi.org/10.5194/amt-12-1183-2019, 2019.

Freud, E., and Rosenfeld, D. (2012), Linear relation between convective cloud drop number concentration and depth for rain initiation, J. Geophys. Res., 117, D02207, doi:10.1029/2011JD016457.

Heymsfield, A. J., Schmitt, C., and Bansemer, A.: Ice cloud particle size distributions and pressure dependent terminal velocities from in situ observations at temperatures from 0 to -86C, J. Atmos. Sci., 70, 4123–4154, 2013.

Hu, Y. X. and Stamnes, K.: An accurate parameterization of the radiative properties of water clouds suitable for use in climate models, J. of Climate, 6, 728–742, 1993.

Yang, P., Bi, L., Baum, B. A., Liou, K.-N., Kattawar, G., and Mishchenko, M.: Spectrally consistent scattering, absorption, and polarization properties of atmospheric ice crystals at wavelengths from 0.2 to 100 mu, J. Atmos. Sci., pp. 330–347, 2013.

Rosenfeld, D., Woodley, W. L., Lerner, A., Kelman, G., and Lindsey, D. T.: Satellite detection of severe convective storms by their retrieved vertical profiles of cloud particle effective radius and thermodynamic phase, Journal of Geophysical Research, 113, D04 208, 2008.

---

## Author Comment (AC1) · 4 Dec 2019

**Response to the reviewers**

We are grateful to the Editor and the two Reviewers for their precious times in reviewing our manuscript. The comments and suggestions of the Reviewers are very helpful and valuable. The issues raised by the reviewers have been addressed (in blue color) in the revised manuscript. Kindly find a point-by-point reply to the issues as follows (presented in blue color).

**Reviewer #1:**

Major points:

1. I strongly suggest that the manuscript present a figure to show how the bi-spectral reflectance vary with cloud optical depth and effective radius. The bi-spectral reflectance algorithm has been widely used (for example, in the cloud retrieval of MODIS). It would be very helpful for the readers to see how sensitive the bi-spectral reflectance is to the cloud optical depth and effective radius for the FY-4 bands. Especially this study shows that the effective radius retrieved from FY-4 is generally smaller than that retrieved from MODIS. I'm curious if this could be caused by some errors in the measurement of the near-infrared reflectance. Of course, the discrepancy between the FY-4 and MODIS retrievals may be caused by other reasons, such as different resolution, different view angles, etc.. I can understand there is discrepancy between instruments. But a figure showing the bi-spectral reflectance as a function of optical depth and effective radius for the FY-4 would be very helpful. The readers could even compare this figure with MODIS.

**Response:** Thanks for your suggestion! We have added a figure to show the look-up table at the specific solar zenith angle, viewing zenith angle and relative azimuth. **[Line 126-141; Figure 3; Table 1]**

In fact, we have also done a comparison with the MODIS look-up table, but due to copyright, this should not appear in the manuscript. The look-up tables are similar overall, and the subtle differences may be due to differences in the center wavelength (FY-4 1.61μm vs MODIS 1.65μm) and spectral response functions.

[Figure]

Figure S1. Bi-spectral solar reflectance lookup table for FY-4 and MODIS. Here, solar zenith angle=20.36°, viewing zenith angle=20.36°, and relative azimuth=0°, with a 0 surface albedo.

2. In Figure 8, I guess it's the median radius that is shown in the figure? I suggest other percentiles, such as the 25 and 75 percentiles, should also be shown in the figure. It seems that the manuscript focuses on some subtle details in Figure 8. For example, it is said in the manuscript that 15-22 microns (03:30-05:30 UTC), correspond to a significantly accelerated growth. But this profile is only the median effective radius. I would like to see if this accelerated growth is still seen in the 25 and 75 percentiles. Similarly, 25 and 75 percentiles should also be plotted in Figure 3.

**Response:** Yes, it is the median radius. We have added relevant descriptions in the figure caption.

Many thanks for your crucial suggestions. The 25 and 75 percentiles have been added in Figure 6 and Figure 8. The two percentiles do provide some valuable information to explain the cloud particle growth. For example, turning points of the 25 and 75 percentiles are at different temperatures. We have added some descriptions. **[Figure 8; Figure 10; Line 292-293]**

3. The discussions related to the microphysical processes on page 8 are not very clear. I don't understand why the cloud in an earlier stage (00:30 UTC) is totally glaciated. The manuscript should at least provide

some description of the convection at this stage. For temperature lower than 273 K, the cloud starts to become mixed phase, so ice-related processes could be very important. But the manuscript seems to emphasize on the collision-coalescence process to explain the accelerated growth for the 15-22 microns (03:30-05:30 UTC). In addition, in lines 250-251, "the rate of increase in Re slows down". Why? I would expect that, in the mixed-phase, the cloud particles are easier to get larger. Line 239-240, "did not exhibit the characteristics of the earlier zones", what does this mean?

**Response:** Sorry for our unclear description. There are three distinct convective cells at 00:30 UTC. According to the maximum temperature gradient method described in the manuscript, the cloud is divided into three cloud clusters. The cloud cluster we are concerned with is small in size and does not contain enough pixels with high temperature (<30 samples in 2.5-K temperature interval). Therefore, we can only see the glaciated part in the profile. **[Line 278-280]**

As pointed out by Reviewer #2, we have completely updated the retrieval algorithm. Based on new results, we have revised the description of Figure 10. Since we are focusing on a convective cloud cluster, some water-phase processes (such as coalescence) also occur above the 0 degree layer. Although cloud particles are easier to get larger in the mixed-phase (increased proportion of ice particles), the increase rate may be slightly lower than the increase rate of coalescence in this case. As reported by Rosenfeld et al. (2014), there are many types of T-Re relations over different regions and different aerosol conditions.

[Figure]

**Fig. 25.** The $T-r_e$ relations of all the types of deep convective cloud in this study.

Rosenfeld, D., et al: High resolution (375 m) cloud microstructure as seen from the NPP/VIIRS Satellite imager, Atmos. Chem. Phys., 14, 2479–2496, doi:10.5194/acp-14-2479-2014, 2014.

Minor points:

(1) In the abstract, lines 13-15: I think these two sentences should be reorganized. Identifying cloud cluster is one task of this study. Obtaining the Re profiles is the other task of this study. The two sentences in the manuscript seem to mix the two tasks together.

**Response:** Thanks! We have split the content into two sentences to make it clear. **[Line 14-15]**

(2) In the introduction: because the Re profile is very useful for studying precipitation formation, the first paragraph of the introduction should have some writings on precipitation. In the first sentence, only the radiation budget is mentioned. I think it should also be pointed out that clouds control precipitation, and therefore the water cycle.

**Response:** Thanks! We have added a reference and some description. **[Line 26-27]**

(3) Line 31: the coalescence of cloud droplets is proportional to Re^5. What does this mean? What property of coalescence is proportional to Re^5?

**Response:** Sorry for the confusing sentence. We have changed it to "that the rate of droplet coalescence is proportional to the 5th power of the mean volume radius" **[Line 32-33]**

(4) Line 76: the aim of this study was to automatically identify and… Therefore I think section 2.2 should be reorganized. The method of identifying the cloud cluster should first be presented, and then the method for obtaining profiles of effective radius is presented. But I'll leave it to the authors to decide on this.

**Response:** Thanks! We have carefully considered this suggestion. Actually, our technical route is, $R_e$ retrieval –> cloud-cluster identification –> $R_e$ profiles. We do not mention $R_e$ retrieval at the aim of study, because the bi-spectral reflectance algorithm has been widely used. Therefore, in the method section, we still maintain the current structure.

In order to make the logic clear, we added subtitles in section 2.2.

(5) In section 2.1: it's better to show the total 14 band wavelength. How many bands are in the visible? How many in the near-infrared? And how many in the infrared

**Response:** Thanks, we agree. We have added a sentence to show the total 14 bands. **[Line 93-95]**

(6) Figure 2: Figure 2 should only plots the north hemisphere. There's no need to plot the southern hemisphere in the figure. The domain shown in Figure 3 should also be indicated in Figure 2. The angles are shown in Figure 2. But I don't see any discussion of the angles in the text.

**Response:** Many thanks for your suggestion. We have revised Figure 2 according to your suggestion. The angles affect the reflectance and are considered in the lookup table. We have added some brief explanation. **[Line 126-127; Figure 2]**

(7) Line 150-152: why is the most extreme precipitation event selected for this study? Please provide some motivation. Intuitively I would expect that this study could start with a normal precipitation case.

**Response:** Thanks. The assumption of the $R_e$-profile retrieval is that $R_e$ and temperature of the cloud top are the same as the $R_e$ and temperature within the cloud at the same height. In order to obtain a complete $R_e$ profile, the distribution of cloud-top temperature should be as wide as possible. Compared to normal stratiform precipitation, the presence of convective precipitation is beneficial for the calculation of the $R_e$ profile. Another reason is that, the Meiyu period was short, the precipitation was weak, and the rain belt was unstable in 2018 (atypical Meiyu year). During the Integrative Monsoon Frontal Rainfall Experiment (June 10[th] to July 10[th]), there were only three precipitation processes. This precipitation case happened to pass through the Xianning Station (114.28E, 29.87N) and the observation data was the most complete. The related results will be published in a JGR special issue (Integrative Monsoon Frontal Rainfall Experiment).

We have added some motivation, and changed the word "extreme" to "heavy". **[Line 189-192]**

**Reviewer #2:**

Major comments:

1. Description of methods

A paper in AMT/ACP should enable the reader to understand and to replicate the presented results and should not limit itself to report on its scientific advancements. The two methods central to this manuscript, however, are not adequately described to guarantee the reproducibility of the presented results. While the description of a bi-spectral retrieval algorithm should be well established and straightforward, the method to find independent cloud clusters seems new and worth to be described in more detail. In the following,

I will try to give more specific advices what is still missing in section "2.2. Methods" and how to organize it in subsections:

**Response:** Many thanks for your important suggestion. We have reorganized the whole section "Methods".

First, I strongly suggest to explain the forward simulations with libRadtran in more detail. The authors should clearly state all cloud parameters and theirs boundaries which have been varied during the forward simulations. To understand the discrepancies between retrieved reff from FY-4 and MODIS, the reader needs to know the range and steps of optical thickness, effective radius, illumination and viewing angles. The current manuscript does not explain how the model clouds were set up, if an standard aerosol environment was considered and if a variable ground albedo was taken into account? The given citation for the used optical properties parameterization for ice clouds (Baum et al., 2014) also does not explain if the baum v36 (Heymsfield et al., 2013; Yang et al., 1993; Baum et al., 2014) parameterization with the general habit mixture, the rough-aggregates or the solid-columns option has been used? Moreover, the authors state that they used the optical properties parameterization for liquid clouds of Hu et al. (1993), but should be aware that the developers of libRadtran state that:

Note that this parameterization has been developed to calculate irradiances, hence it is less suitable for radiances. This is due to the use of the Henyey-Greenstein phase function as an approximation of the real Mie phase function.

**Response:** Thank you very much for your important suggestions.

1. We have added a table to show the grid point values of the lookup table, including $\tau$, $R_e$, and angles. **[Table 1]**

2. An example of $\tau$ and $R_e$ lookup table over ocean_water underlying surface at specific angles has been added. The comparison with MODIS lookup table could be seen in Figure S1. **[Figure 3]**

3. In order to simplify the model and speed up the calculation, we closed the aerosol module, and only set the two types of underlying surfaces, mixed_forest and ocean_water. **[Line 121-124]**

4. We have completely recalculated the lookup table. Here, we adopt rough-aggregate option for ice cloud, and Mie scheme for water clouds. **[Line 118-120]**

As a result, the others figures have changed slightly.

Second, the authors do not describe how they handle the phase discrimination between water and ice clouds at all. The example discussed in section 4 clearly contains water as well as ice clouds. Inferring from the retrieved reff, the retrieval seems to handle water and ice clouds quite well. There is, however, no explanation if a threshold technique is used to separate ice from water clouds and how the mixed-phase region is handled. This is especially important since the discussion in section 4 is focused on the region of cloud glaciation.

**Response:** Many thanks for your crucial suggestion. There are many possibilities in thermal infrared pixels with 4 km resolution, so we refer to the MODIS algorithms, both using the threshold method and interpolation method.

Briefly, 273 K and 233 K are used as thresholds for pure water clouds and pure ice clouds. When the bright temperature is between 233 K and 273 K, we bring the reflectance into the water-cloud and ice-cloud lookup table simultaneously. If only one $R_e$ is successfully retrieved, use this value, otherwise perform linear interpolation based on the differences between bright temperature and 233 K (273 K). **[Line 129-141]**

At last, section 2.2 "Methods" also introduces the technique to identify cloud clusters which is a central aspect of this study. While this technique should get its own subsection, it also deserves a more visual and complete description: The authors miss to provide important details, like the size of the Gaussian filter and the used value of the distance threshold. Furthermore, the authors write that "the local temperature minimum" (P5, L135) is determined, but to the reader it is not clear if this is done pixel-wise or for the complete scene. Do you identify all local temperature minima in the scene or only for a search radius around each pixel?

Moreover, the authors write on P5, L139ff:

"3) Combining the processed 10.8 micron brightness temperature and the local minimum using the maximum temperature gradient method, a sequential search is carried out to determine the convective core to which each pixel belongs, thereby dividing the cloud clusters."

Here, it is not clear how the maximum temperature gradient method (which is never explained!) can be combined with a brightness temperature to determine the convective core for each pixel in a sequential search (which is also not explained). Here, a descriptive figure could significantly improve the comprehensibility of this paragraph. In my opinion, the revision of this section should be of major

concern since it seems to be the main novelty of this work.

**Response:** Sorry for our unclear description.

1. We have added subtitles for $R_e$ retrieval and Cloud-cluster identification. Some important details have been provided. **[Line 143; Line 160-161]**

2. We have added some details for local temperature minimum. We firstly identify all local temperature minima in the entire scene, and then merge the minimums within the distance threshold (40 km). **[Line 168-170]**

3. We have added a schematic diagram for the maximum temperature gradient method. We have also provided the specific calculation methods. **[Line 173-178; Figure 4]**

2. Discussion of results

As also pointed out by RC1, the discussion about the microphysical evolution of the cloud cluster on page 8 is not very convincing. In my opinion, the authors focus on details in Figure 8 and on processes (collision-coalescence, precipitation formation), which their spaceborne technique probably never can resolve in detail. As long as the handling or the influence of mixed-phase cloud regions is not explained, their discussion oversells their approach while it misses to highlight its strength: to observe the timescales between initiation, invigoration and the mature phase of a convective cluster.

**Response:** Many thanks for your constructive suggestions. We agree that our current technique may not be able to reveal some subtle details of the precipitation process perfectly due to the coarse resolution of the instrument and the uncertainty in precipitation itself. We have removed some over-explained texts (or speculation). Thank you for recognizing our technique of objective cloud cluster identification. We have added some description of cloud-cluster morphology.

Examples of unclear or unproven sentences are:

Page 8, L239f: "Re changed almost linearly with temperature between 285 and 230 K and did not exhibit the characteristics of the earlier zones" What do you mean here by "earlier zones"?

**Response:** We have revised it to "did not exhibit a clear boundary between the diffusional growth zone, the coalescence growth zone, the rainout zone and the mixed phase zone". **[Line 284-285]**

Page 8, L241f: "Under the influence of such strong ascending motion, the boundary between the zones

is broken and there is not enough time for the growth of precipitation."

This statement is incomprehensible to me since I can not observe clear boundaries in Figure 8.

**Response:** Sorry for our unclear description. When the system is mature, the boundaries of the zone would be clear because of the differences in different physical processes, such as Figure 10e and Figure 10f. In the early stage (such as Figure 10b), the boundary is broken, and $R_e$ changed almost linearly because of the strong ascending motion. **[Line 286]**

Page 8, L255f: "In addition, because of the deposition of aerosols after precipitation, sufficient water vapor allowed Re to exceed 20m at higher temperatures." Can you deduce this observation from your retrieval results alone? I doubt that you can observe the deposition of aerosols from a geostationary satellite. As the cluster during this phase is mainly governed by a thinning anvil, multilayer cloud effects have to be taken into account for the discussion of the observed reff profile.

**Response:** It is our speculation instead of observation. Based on current data, we cannot retrieve aerosols under clouds. This sentence has been deleted.

We totally agree with you that multilayer cloud and anvil dominate, which affects the retrieval of $R_e$ profile. Thanks again for your reasonable explanation! We have added some discussion. **[Line 299-202]**

Minor comments

Title: I suggest a slight change to the title of the manuscript, since the original title "Retrieval of the vertical profile ... " gives the impression of a retrieval which can be applied to a single cloud like multi-wavelength retrievals (e.g. Chang et al. (2003)). In my opinion, the title "Retrieval of the vertical evolution of the cloud effective radius from the Chinese FY-4 next-generation geostationary satellite" better captures the approach to retrieve the vertical profile of reff by observing the evolution of reff using a cloud ensemble approach.

**Response:** We appreciate your nice suggestion. The title has been revised.

References: Please check all references for chronological order

**Response:** We have checked all references, and they are arranged in chronological order now.

P1, L31: Freud and Rosenfeld (2003) showed that the rate of droplet coalescence is proportional to the

mean volume radius $Rv^5$ and not the mean effective radius $Reff^5$.

**Response:** Thanks. We have changed it to "that the rate of droplet coalescence is proportional to the 5th power of the mean volume radius" **[Line 33-34]**

P2, L38f: Reword you sentence "More aerosols result in more cloud condensation nuclei (CCN), leading to a higher height of the 14um threshold for Re and a smaller coalescence efficiency" into "More aerosols result in more cloud condensation nuclei (CCN) and smaller reff with coalescence occurring at an higher altitude during ascent"

**Response:** Thanks. We have changed this sentence according to your nice suggestion. **[Line 40-41]**

P2, L49f: Besides the multi-wavelength approach you should also mention the cloud side perspective approach to directly retrieve the vertical profile of reff (e.g. Ewald et al. (2019)).

**Response:** Many thanks for your nice suggestion and the important reference. We have mentioned this approach in the revised manuscript. **[Line 56-58]**

P3, L74f: "To the best of our knowledge, no instrument has yet provided an official Re vertical profile product." This statement is not true. You should at least mention multi-instrument products like DARDAR, 2C-ICE or Cloudnet which provide effective radius profiles on an operational basis:

Delanoe, J., and R. J. Hogan, 2010: Combined CloudSat-CALIPSO-MODIS retrievals of the properties of ice clouds. J. Geophys. Res., 115, D00H29.

Deng M, Gerald G. Mace, Zhien Wang, and R. Paul Lawson, 2013: Evaluation of Several A-Train Ice Cloud Retrieval Products with In Situ Measurements Collected during the SPARTICUS Campaign. J. Appl. Meteor. Climatol., 52, 1014–1030.

**Response:** Sorry for our misleading statement. We have revised the description. **[Line 78-80]**

Moreover, it is not clear what you mean with "official". Maybe you mean "operational" in this context?

**Response:** Yes. Changed. **[Line 80]**

• P4, L95 "We selected Chinese regional data". Please be more precise what data and from which source (model, measurements?).

**Response:** Sorry for our unclear description. FY-4 AGRI has two scanning methods: full-disk scan and

Chinese regional scan. At 4 km resolution, full-disk scan has 2748*2748 pixels, and Chinese regional scan has 1108*2748 pixels. Here, we selected FY-4 AGRI Chinese regional scan data. We have added some descriptions. **[Line 96; Line 101]**

P4, L98 Please refer to different sub-panels (a, b, c) in Figure 2. Moreover I do not understand what you mean with "closely related to the retrieval".

**Response:** We have reorganized this sentence and added specific names of channels. **[Line 103-105]**

P4, L100 "The spectral retrieval algorithm of cloud properties is based on the characteristics of the cloud itself and the bi-spectral reflectance algorithm is the most representative."
This sentence is incomprehensible. What do you mean by that?

**Response:** Sorry for this unclear sentence. We have revised it to "the spectral Re retrieval algorithms, in which the bi-spectral reflectance algorithm is the most representative, are based on the optical characteristics of the cloud itself." **[Line 108-109]**

P5, L130 Please elaborate what you mean by "The original data are preprocessed".

**Response:** Sorry for this unclear description. "Preprocess" and "passed through a Gaussian filter" mean the same. We have revised it to "the 10.8 μm channel brightness temperature is pre-processed through a Gaussian filter". **[Line 160]**

P5, L154ff You only mention MODIS in your manuscript and never mention the satellite and the actual retrieval product you used in your comparison. You obviously used data from MODIS on Terra. Moreover, there are multiple scientific datasets (SDS) for cloud effective radius retrieved with different techniques and filters. Did you use the SDS Cloud_Effective_Radius_16 with the same channel combination?

**Response:** Many thanks for your crucial suggestions. We used the variable Cloud_Effective_Radius in the first draft. We have already recalculated the figures and used Cloud_Effective_Radius_16 (Cloud_Optical_Thickness_16). **[Line 193; Line 200; Line 493]**

P6, L174 Weather radars (here with a coarser resolution than the satellite!) should not be used to explain resolution effects between different satellites working in the visible wavelength region. Drizzle or a few

rain drops in a pixel can give you a radar signal which seems to be clear in the visible wavelength region.

**Response:** Sorry for our unclear description. Here, precipitation radar (PR) and visible and infrared scanner (VIRS) are both onboard the same satellite (TRMM, Tropical Rainfall Measuring Mission satellite). Many studies merge the two to analyze "warm rain" or the others. We agree that many possibilities exist in the pixel (especially different resolution pixel). **[Line 214-215]**

P9, L274 "The glaciation temperature increased significantly during the period of dissipation" Have you shown this observation in the results? And with which method?

**Response:** Sorry for this misleading sentence. We have deleted it in the revised manuscript.

Wording

P1, L2 "from the first of the Chinese" ... "from the first Chinese"

**Response:** Thanks. Changed. **[Line 13]**

P1, L28 "determining the effects of radiation and the water cycle on the Earth's climate system" ... "determining their impact on the water cycle and their radiative effects on Earth's climate system"

**Response:** Thanks. Changed. **[Line 31]**

P2, L55 "obtain" ... "correlate"

**Response:** Thanks. Changed. **[Line 59]**

P3, L93 "the shortwave distribution of AGRI" ... "the shortwave spectral characteristics of AGRI bands"

**Response:** Thanks. Changed. **[Line 99]**

P4, L97 Please rephrase "... and the three angles important for retrieval"

**Response:** Thanks. This sentence has been reorganized. **[Line 104]**

P9, L259 "We used bi-spectral reflectance observations from the FY-4 AGRI to calculate a lookup table to retrieve Re and $\tau$ ." This sentence does not make sense.

**Response:** Thanks. This sentence has been reorganized. **[Line 305]**

[revised manuscript text omitted]

---

## Author Response (AR2)

**Response to the reviewers**

We are grateful to the Editor and the two Reviewers for their precious times in reviewing our manuscript. The issues raised by the reviewers have been addressed (in blue color) in the revised manuscript. Kindly find a point-by-point reply to the issues as follows (presented in blue color).

Referee #1

I think that the manuscript now qualifies for publication in ACP. The data and method are described pretty well. The Re profiles from 00:30 UTC to 01:30, 02:30, 03:30, 04:30, and 05:30 are now discussed in more detail. The manuscript is in good quality overall. I would suggest it be accepted for publication in ACP. Here are some minor suggestions:

Line 140: "mixed-cloud" should be "mixed-phase cloud"?

**Response:** Thanks! Changed. [Line 140]

Line 288: "the regional difference" is not very good. It could be changed to "the multiple zones"?

**Response:** Thanks! Changed. [Line 291]

Line 292: Why do the 25 and 75 percentiles show different turning points of Re and growth rates from the median? Are there any explanations?

**Response:** Turning points and growth rates are affected not only by temperature, but also by *Re*. The 25th and 75th percentiles have different sizes of *Re*, so turning points and growth rates will also be different. [Line 296-297]

Referee #2

1) Can the authors check if their retrieval is influenced by the absorption lines of CO2 within the 1.61 micron channel of FY-4A? Looing at the atmospheric transmittance between the surface and high ice clouds (18 km) in Figure 1, there might be some cloud height depedence on the retrieval to which the MODIS channel at 1.65 micron is not sensitive to? A simple radiative transfer calculation should be sufficient and a short sentence in the manuscript if this needs to be considered.

**Response:** Many thanks for your important suggestion! The retrieval is indeed affected by factors such as water vapour and CO2. We conducted the tests under the most extreme conditions, that is, setting the ice cloud top to 0 km and 18 km respectively. The results show that the 1.61 μm reflectance in the two extreme cases differ by ~8%, which is acceptable. In other words, the impact of cloud height difference in reflectance will not exceed 8%. [Line 141-144]

[Figure]

Figure S1. Bi-spectral solar reflectance lookup table for FY-4. (a) Ice cloud at 0 km; (b) Ice cloud at 18 km.

2) L160-L161:

"... standard deviation of 10 and truncated at 4 times the standard deviation ..."

A unit is still needed here: 10 pixel or 10 km?

**Response:** It is 10 pixels. [Line 164]

3) L168

"... we first calculate the local temperature minimum of the complete scene ..." -> "... we first calculate local temperature minima for the complete scene ..."

**Response:** Thanks. Changed. [Line 171]

4) Figure (3) 1.65 -> 1.61

**Response:** Thanks. Changed. [Figure 3]